# Observation of Cu Spin Fluctuations in High-*T*_c_ Cuprate Superconductor Nanoparticles Investigated by Muon Spin Relaxation

**DOI:** 10.3390/nano11123450

**Published:** 2021-12-20

**Authors:** Suci Winarsih, Faisal Budiman, Hirofumi Tanaka, Tadashi Adachi, Akihiro Koda, Yoichi Horibe, Budhy Kurniawan, Isao Watanabe, Risdiana Risdiana

**Affiliations:** 1Department of Physics, Padjadjaran University, Sumedang 45363, Indonesia; 2Department of Human Intelligence Systems, Kyushu Institute of Technology, Kitakyushu 808-0196, Japan; faisalbudiman@telkomuniversity.ac.id (F.B.); tanaka@brain.kyutech.ac.jp (H.T.); 3Research Center for Neuromorphic AI Hardware, Kyushu Institute of Technology, Kitakyushu 808-0196, Japan; horibe@post.matsc.kyutech.ac.jp; 4Department of Engineering and Applied Sciences, Sophia University, Tokyo 102-8554, Japan; t-adachi@sophia.ac.jp; 5Muon Science Laboratory, KEK, Ibaraki 305-0801, Japan; coda@post.kek.jp; 6Department of Material Science and Engineering, Kyushu Institute of Technology, Kitakyushu 804-8550, Japan; 7Department of Physics, Universitas Indonesia, Depok 16424, Indonesia; budhy.kurniawan@sci.ui.ac.id; 8Meson Science Laboratory, RIKEN, Saitama 351-0198, Japan

**Keywords:** Cu spin fluctuations, high-*T*_c_ cuprate superconductor, magnetic correlation, muon-spin relaxation, nano-size effects

## Abstract

The nano-size effects of high-*T*_c_ cuprate superconductor La_2__−*x*_Sr*_x_*CuO_4_ with *x* = 0.20 are investigated using X-ray diffractometry, Transmission electron microscopy, and muon-spin relaxation (*μ*SR). It is investigated whether an increase in the bond distance of Cu and O atoms in the conducting layer compared to those of the bulk state might affect its physical and magnetic properties. The *μ*SR measurements revealed the slowing down of Cu spin fluctuations in La_2__−*x*_Sr*_x_*CuO_4_ nanoparticles, indicating the development of a magnetic correlation at low temperatures. The magnetic correlation strengthens as the particle size reduces. This significantly differs from those observed in the bulk form, which show a superconducting state below *T*_c_. It is indicated that reducing the particle size of La_2__−*x*_Sr*_x_*CuO_4_ down to nanometer size causes the appearance of magnetism. The magnetism enhances with decreasing particle size.

## 1. Introduction

Nanoparticles have gained intense interest recently in terms of their potential applications and fundamental physics. For applications, it is reported that nanoparticles have the potential to be used to improve the performance of gas sensors [1], catalysts [2], biomedicine [3], high-density magnetic storage [4], and magnetic fluids [5]. From a fundamental physics viewpoint, the physical properties alter significantly from those observed in bulk materials when the particle size of the material is reduced down to the nanometer size [6,7,8,9,10]. As an example, ferromagnetic behavior was observed in gold which has been considered as non-magnetic. This occurred when the size of the gold cluster was about 2.2 nm [10,11]. Even though magnetism was observed, the question remains in this study whether the magnetism comes from the encapsulated molecules or from inside the gold nanoparticles.

An antiferromagnet CuO was also reported to behave ferromagnetically, showing a hysteresis loop at 8 K and 300 K when reducing the particle size to 25 nm [12,13,14]. In addition, the magnetic transition temperature, *T*_N_, was reported to be drastically decreased to 12 K from that observed in the bulk form of about 229 K when reducing the particle size down to 2–3 nm. The surface effect was discussed to explain the presence of ferromagnetic components and the decrease in *T*_N_ [7,12]. 

An observation of magnetism was not only reported in non-magnetic and antiferromagnetic but was also reported to appear in the high-*T*_c_ cuprate superconductor (HTSC) which is La_2__−*x*_Sr*_x_*CuO_4_. An anomaly upturn of magnetic susceptibility was observed below the critical temperature, *T*_c_, with Sr concentration from 0.10 to 0.20, and the particle size was about 113 nm [15]. This report raises new questions since it differs significantly from those reported in bulk cases. In the bulk case, it behaves superconducting below *T*_c_ without the trace of magnetism that can be observed at 0.05 < *x* < 0.25 [16,17,18,19,20]. In addition, the magnetic ordering was considered to explain the origin of the 1/8 anomaly in the bulk HTSC. The 1/8 anomaly is a phenomenon where the *T*_c_ is suppressed at *x* = 1/8. The suppression of *T*_c_ at *x* = 1/8 was proposed due to the incommensurate magnetic ordering [20,21,22,23,24]. This was linked to the stripe theory in which the spin domain and hole domain are self-organized in the CuO_2_ plane [20,22,24]. In the case of ferromagnetic spin fluctuation, it was predicted to occur in a highly over-doped regime at *x* > 0.25 causing the suppression of superconductivity in this regime [25,26]. The reason for the observation of magnetism in nano-HTSC at a wide range of doping concentrations is unclear, including whether the surface effect or stripe theory plays an important role. Moreover, prior studies of nano-size effects in La_2__−*x*_Sr*_x_*CuO_4_ at *x* = 0.10–0.30 used samples with particle sizes larger than 100 nm which are not appropriate to reveal nano-size effects. Consequently, how the nano-size affects La_2__−*x*_Sr*_x_*CuO_4_ is still an open question.

To obtain a sample with a particle size of less than 100 nm, the chemical reaction method sol–gel was applied to control its particle size. There are many fabrication methods to prepare nanoparticles, either top-down or bottom-up methods. However, the chemical reaction method sol–gel is frequently used [27,28]. In this study, we prepared free-standing nanoparticles without any capped molecules to eliminate the effect from the molecule, which might provide magnetism. The particle size was controlled to be lower than 100 nm.

The Sr concentration of 0.20 was chosen for the study to eliminate the possibility of the largening of the density of state at the Fermi level, which caused the enhancement of ferromagnetic fluctuation at the highly over-doped regime [25,26,29]. Thus, we can focus on the magnetism that might be observed under nano-size effects. In addition, superconductivity is set to occur at *x* = 0.20 [16,17] allowing physical properties between bulk and nano-sample to be compared. Muon-spin relaxation, *μ*SR, was chosen to investigate the magnetic state of La_2__−*x*_Sr*_x_*CuO_4_ nanoparticles. It is because *μ*SR is a sensitive probe that it can easily detect magnetic ordering and slowly fluctuating magnetism [26].

## 2. Materials and Methods

La_2__−*x*_Sr*_x_*CuO_4_ nanoparticles were synthesized with an *x* = 0.20 powder sample using the sol–gel method. The starting precursors were La_2_O_3_ (Aldrich Chemical Co., Inc., Milwaukee, Wisconsin, USA, 99.9%), SrCO_3_ (Aldrich Chemical Co., Inc., Milwaukee, WI, USA, 99.9%), and CuO (Wako Pure Chemical Industries, Ltd., Osaka, Japan, 99.9%). The La_2_O_3_ was pre-fired at 973 K for 2 h to remove air and water components that might exist. After that, La_2_O_3_, SrCO_3_, and CuO were weighed following the stoichiometric calculation. La_2_O_3_ and CuO were dissolved with a nitric acid solution (Wako Pure Chemical Industries, Ltd., Osaka, Japan, concentration 70%) separately. After a transparent solution of La(NO_3_)_3_ and Cu(NO_3_)_2_ was obtained, they were mixed into one solution. Then, SrCO_3_ was added to the solution and stirred until a homogeneous solution was achieved. Citric acid monohydrate (Wako Pure Chemical Industries, Ltd., Osaka, Japan, 99.5%) and ethylene glycol (Wako Pure Chemical Industries, Ltd., Osaka, Japan, 99.5%) were added to the solution to control the transformation process from a solution to a gel and to fasten the binding reaction between La^3+^, Sr^2+^, Cu^2+^, and O^2-^. An ammonia solution (Wako Pure Chemical Industries, Ltd., Osaka, Japan, concentration 28%) was also added to the solution to adjust the pH level. This is an important process because citric acid monohydrate will be active at a pH of 7.0. After the gel was formed, all the water and organic components were removed by heating it at 423 K and 573 K, respectively. A sponge-like product was obtained after this process. The product was grounded to form a powder. Then, the powder was sintered for the nucleation process growing the La_2__−*x*_Sr*_x_*CuO_4_ phase. The sintering temperature and time were set to control the particle size. The final product was in powder form. Two samples were prepared for the present *μ*SR measurements with two different sintering conditions, which are 973 K for 180 min and 1073 K for 180 min.

X-ray Diffractometry (XRD) measurements with Cu-K*α* radiation using Rigaku MiniFlex 600, Tokyo, Japan, were performed at room temperature to check the quality of the samples. The crystal structure, lattice parameters, and crystallite size were analyzed by Rietveld refinement of the XRD patterns using the GSAS II package developed at Advanced Photon Source, Argonne National Laboratory, IL, USA [30]. Transmission electron microscopy (TEM) was carried out at room temperature to examine the morphology and particle size of the samples by using STEM HITACHI HD-2000, Tokyo, Japan, operated at 200 kV. To minimize the agglomeration that might occur during the measurement, a sample of about 10 mg was dropped into 5 mL of ethanol. After that, it was dispersed in an ultrasonic cleaner for five minutes. A drop of the sample solution was put on a grid. The grid size was 200 mesh and the grid diameter was 3.05 mm. The carbon membrane was placed on the top of the grid to fix the position of the sample. Then it was dried overnight in a vacuum desiccator. After that, it was set in the sample chamber of the TEM system. TEM analyses were performed by taking 30 acquisitions from different areas of the sample to determine the particle size distribution. The ImageJ (Java-based, free software) developed at the National Institutes of Health and the Laboratory for Optical and Computational Instrumentation, University of Wisconsin, Wisconsin, USA, was used to measure the particle size of each sample.

The *μ*SR measurements were conducted in the zero-field (ZF) condition at the Materials and Life Science Experimental Facility of the J-PARC, Japan, under a user program (Proposal No. 2018B0322 and 2019B0413) using a pulsed-positive-surface muon beam. The powder sample was packed in two pieces of high-purity silver (Ag) foil (99.97%). A spatial fly-path set-up using an extended vacuum chamber was installed along the beamline to reduce the background of muon signals. Muons that did not penetrate the sample will pass through along the extended vacuum chamber then stop at the end of this chamber, placed at a considerable distance from the sample position and detectors. Using this fly-path set-up, the background signal can be reduced up to 25% more than the usual set-up [31]. The asymmetry parameter, *A*(*t*), of muon-spin polarization at a time *t* is defined in Equation (1).
(1)$$A\left(t\right)=\frac{F\left(t\right)-\alpha B\left(t\right)}{F\left(t\right)+\alpha B\left(t\right)}$$
where *F*(*t*) and *B*(*t*) are the total number of muons detected by forward and backward counters aligned along the beamline, respectively. *α* reflects the different efficiency between these counters. The *μ*SR time spectra, namely, the time evolution of *A*(*t*) was measured from 50 K down to 3.3 K with 3 K increment step to detect the appearance of a magnetic ordering and/or fluctuating magnetism.

## 3. Results and Discussion

The X-ray diffraction patterns of La_2__−*x*_Sr*_x_*CuO_4_ with *x* = 0.20 are depicted in Figure 1. The red line, dark-blue line, light-green line, and cyan bar represent the observed data, the calculated patterns, the difference between the observed data and the calculated data, and the peaks of La_2__−*x*_Sr*_x_*CuO_4_, respectively. The difference between the observed and calculated data is correlated to the goodness of fit (GOF) showing the refinement quality. The XRD patterns are in good agreement with the XRD pattern of bulk La_2__−*x*_Sr*_x_*CuO_4_ with *x* = 0.20 (COD with ID 1008481). The major peaks of the samples are assigned to the La_2__−*x*_Sr*_x_*CuO_4_ phase. No impurity peak can be detected, indicating the samples are single phase. Rietveld refinement analysis confirmed that all samples have tetragonal symmetry with the space group of *I4/mmm*. The GOF value of each sample is less than 1.50 from which it is inferred that the refinement process is good since the GOF value should be around 1.0. Table 1 and Table 2 summarize the Rietveld refinement results and atomic coordinates, respectively. The volume of the unit cell tends to increase when lowering and shortening the sintering temperature and time. The schematic figure of the sample’s crystal structure is displayed in the inset of Figure 1b. The La atom, Sr atom, Cu atom, and O atom are represented by black, yellow, brown, and magenta balls, respectively. O1 and O2 are the so-called planar and apical oxygen, respectively.

The average crystallite size was determined by using Scherrer’s equation given in Equation (2), where *K* = 0.9 is a constant corresponding to the crystallite factor, *λ* is the wavelength of the XRD, *D*_s_ is crystallite size, *β*_hkl_ is full width at half maximum (FWHM) of the strongest peak intensity, and *θ* is Bragg angle. The average crystallite size for the two samples is 64 and 39 nm.
(2)$$D_{s}=\frac{K\lambda }{\beta_{hkl}cos\theta }$$

The selected bond distances are given in Table 3. The bond distances of the bulk sample are also presented in the table for comparison [32]. It shows that particle size reduction causes changes in the bond distances. Compared to the bulk state, the most significant changes are those found in the bond distance of Cu1–O2 and Cu1–O1, indicating a difference in the size of the conducting layer. An increase in the size of the conducting layer derived from particle size reduction to nanometer size might have direct implications for the various properties of this material, both its electrical [33] and magnetic properties [6,7,12,15].

Figure 2 shows the TEM image and particle size distribution of La_2__−*x*_Sr*_x_*CuO_4_ nanoparticles with *x* = 0.20. The solid line in Figure 2b,d displays the log-normal size distribution fitting result. In the case of the sample with the sintering condition of 1073 K for 180 min, the particle size is distributed from 14.77 to 129.14 nm, with a mean size of 73.08 nm. For the sample with the sintering condition of 973 K for 180 min, the particle size is distributed from 12.65 nm to 104.50 nm, with a mean size of 37.33 nm. The particle size obtained from TEM measurements gave comparable results to those estimated from XRD measurements with a different value of about 4–14%. It is in line with that reported in nano-CoPt_3_. The particle size measurements using XRD showed a good agreement within 5–7% with the values measured by TEM [34].

The agglomeration was still observed in the TEM results even though preventive measures have been taken by dissolving the powder sample with ethanol and dispersing it in an ultrasonic cleaner. In the case of XRD, the whole area of the sample was measured, and all of the XRD peaks were also included in the calculation of the particle size. It is indicated that the particle size obtained from XRD shows the appropriate average particle size. Hence, the particle size of the sample is determined from that obtained by XRD. From these results, it can be seen that the higher the sintering temperature and the longer the sintering time, the bigger the resulting particle size. It is inferred that the particle size is tunable and strongly depends on the sintering condition. The detailed sample preparation has been reported in our previous paper [35].

Figure 3a,b shows the time dependence of asymmetry for La_2__−*x*_Sr*_x_*CuO_4_ with *x* = 0.20 with a particle size of 64 nm and 39 nm, respectively. To ease the reading of the graph, we show asymmetry at 3.3 K, 6 K, 15 K, and 50 K. The time spectra were gradually changed from Gaussian to exponential behavior by decreasing the temperature in both samples. The Gaussian behavior is represented by the time spectra at 3.3 K, while exponential behavior is represented by the time spectra at 50 K. Both time spectra are shown in the inset of Figure 3a. In the bulk state, it is reported that the time spectra did not change from Gaussian to exponential even down to 0.3 K [17]. The change of the time spectra from Gaussian to exponential with decreasing temperature in La_2__−*x*_Sr*_x_*CuO_4_ nanoparticles with *x* = 0.20 indicates the slowing down of Cu spin fluctuations. The slowing down of Cu spin fluctuation reveals the magnetic correlation that was developed at low temperatures. In addition, there is no muon-spin precession in both samples down to 3.3 K, indicating the absence of long-range magnetic ordering. It is the addition of Sr to the La site which causes the disappearance of long-range magnetic ordering as reported in the bulk state [16,17,18,19]. However, in the bulk state, the slowing down of Cu spin fluctuations at low temperatures was not observed [17].

The ZF-*μ*SR time spectra of La_2__−*x*_Sr*_x_*CuO_4_ nanoparticles with *x* = 0.20 were analyzed using Equation (3). Solid lines in Figure 3a,b show the best-fitting results.
(3)$$A\left(t\right)=A_{0}e^{-\lambda t}G_{z}\left(\Delta ,t\right)$$

*A*_0_ is initial asymmetry at *t* = 0, *λ* is muon-spin relaxation rate, *G_z_*(Δ,*t*) is the static Kubo-Toyabe function expressed by Equation (4).
(4)$$G_{z}\left(\Delta ,t\right)=\frac{1}{3}+\frac{2}{3}\left(1-\Delta^{2}t^{2}\right)e^{-\frac{1}{2}\Delta^{2}t^{2}}$$
where Δ is the half-width of the static internal magnetic field distributed at the muon site.

Figure 3c displays the ZF-*μ*SR time spectra at the lowest temperature of around 3 K for both samples. It can be seen that in the 39 nm sample, the muon-spins decouple slower than in the 64 nm sample, indicating that the development of magnetic correlation gets stronger as particle size reduces.

The temperature dependence of the muon-spin relaxation rate for La_2__−*x*_Sr*_x_*CuO_4_ nanoparticles with *x* = 0.20 is depicted in Figure 3d. Solid lines are to guide the reader’s eye. For the 64 nm sample, the muon-spin relaxation rate starts to increase below 30 K then rapidly increases below 10 K, which is lower than the *T*_c_ of the bulk La_2__−*x*_Sr*_x_*CuO_4_ with *x* = 0.20 of about 27 K [16,17]. It is inferred that the fluctuations of Cu spins slow down with decreasing temperatures, exhibiting the development of magnetic correlation at low temperatures. The muon-spin relaxation rate of the 39 nm sample is higher than that of the 64 nm sample. This result is in line with the result in Figure 3c, which shows that the decoupling of muon-spins is lowered in the 39 nm sample. These results reveal magnetic correlation, which is more developed in the sample that has a smaller particle size.

Referring to the magnetization results of La_2__−*x*_Sr*_x_*CuO_4_ with *x* = 0.15, the superconducting state was reported to be observed when the particle size was about 748 nm [15]. The *T*_c_ of about 36 K is the same as the *T*_c_ of the bulk sample [17]. The superconductivity disappeared, showing Curie behavior when the particle size was reduced to 66 nm. This indicates that the superconducting state was strongly suppressed due to particle size reduction down to nanometer size [15]. The emergence of Cu spin fluctuations and the suppression of the superconducting state in La_2__−*x*_Sr*_x_*CuO_4_ nanoparticles shows that reducing the particle size caused the appearance of weak magnetism.

There are some suggested possible reasons to explain the existence of weak magnetism in La_2__−*x*_Sr*_x_*CuO_4_ nanoparticles. First, the disruption of the lattice periodicity indicated by an increase in the bond distance of Cu1–O1 and Cu1–O2 leads to an increase in the size of the conducting layer. The increase in the size of conducting layer may cause the charge redistribution that might affect the exchange interaction between Cu and O. As reported in CuO with the particle size of 8.8 nm, the appearance of magnetism could be linked to the charge redistribution in both the 4s and 3d electrons of Cu ion. It is proposed that the charge redistribution was triggered by the lattice periodicity disruption at the surface [14]. Further theoretical investigation on the distribution of the charge carrier in HTSC nanoparticles is needed, especially the electronic charge of Cu and O ions.

Another possibility is that the nano-size effects might cause the dense distribution of nanoscale pinning sites, leading to the observation of magnetism. In the case of GdBaCuO nanotapes, there was an unexpected broadening of vortex profiles typically with an FWHM of 6 μm and magnetization loops observed from 10 K up to 83 K [36]. These behaviors are different from those observed in bulk GdBaCuO which show a superconducting state with a *T*_c_ of about 94.5 K [37]. Therefore, it is suggested that the broadening of the vortex profile and the appearance of magnetization loops is due to nanoscale pinning sites [36]. Considering the GdBaCuO nanotapes studies, the observation of magnetism in La_2__−*x*_Sr*_x_*CuO_4_ nanoparticles might be because particle size reduction induced the pinning sites. The pinning sites lead to the development of magnetic correlation at low temperatures. If this is the case, nano-size effects might play the same role as the impurities effect at the Cu site in the bulk state. Impurity substitution at the Cu site is reported to cause the appearance of short-range magnetic ordering and suppression of superconductivity [17,38,39], which is also similar to the impact caused by reducing the particle size down to nanometer size.

## 4. Conclusions

We have investigated nano-size effects in La_2__−*x*_Sr*_x_*CuO_4_ with *x* = 0.20 by XRD, TEM, and *μ*SR measurements. The XRD measurements revealed a change in the size of the conducting layer indicated by a change in the bond distance between Cu and both planar and apical oxygen with reduced particle size. From the *μ*SR measurements, it has been found that the time spectra change from Gaussian to exponential behavior with decreasing temperature exhibits the slowing down of Cu spin fluctuations. The relaxation rate of muon-spin rapidly increases below about 10 K, indicating magnetic correlation development at low temperatures. The muon-spin relaxation rate increases in smaller particle sizes showing that magnetic correlation is more developed with reduced particle size. The present studies show that nano-size effects cause the appearance of magnetism in La_2__−*x*_Sr*_x_*CuO_4_ which enhances with decreasing particle size.

## Figures and Tables

**Figure 1 nanomaterials-11-03450-f001:**
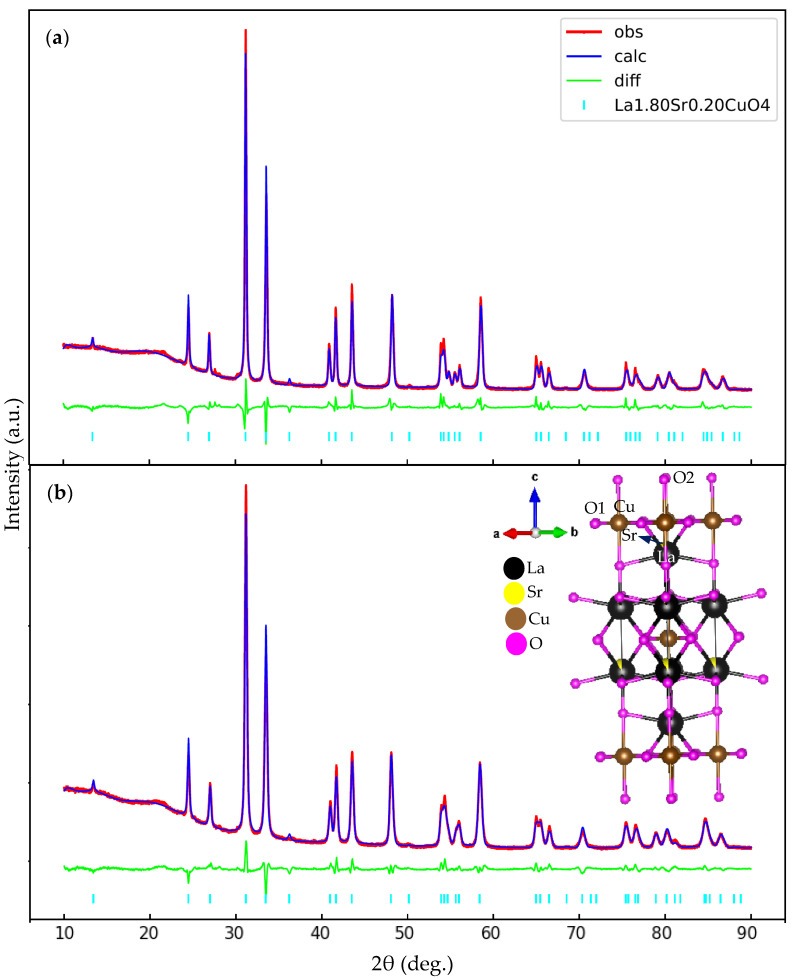
Room temperature of XRD patterns of La_2__−*x*_Sr*_x_*CuO_4_ nanoparticles with *x* = 0.20 under different sintering conditions (**a**) sintering condition of 1073 K for 180 min (**b**) sintering condition of 973 K for 180 min. The inset depicts the schematic figure of the crystal structure of the sample.

**Figure 2 nanomaterials-11-03450-f002:**
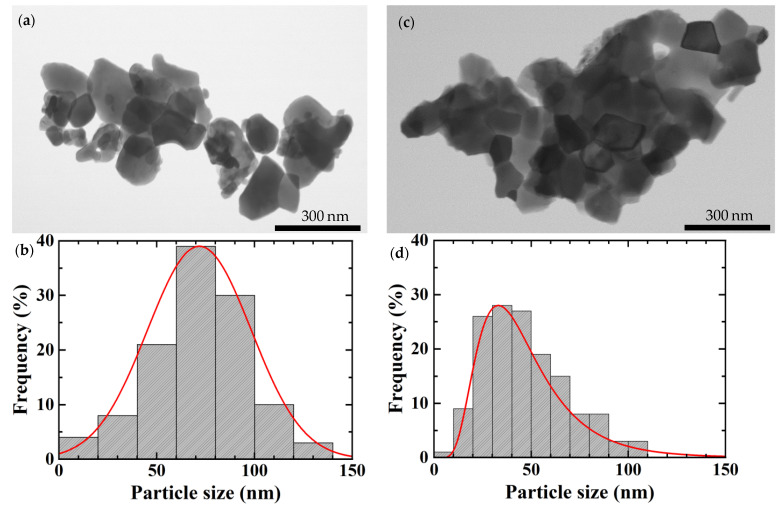
TEM image and particle size distribution of La_2__−*x*_Sr*_x_*CuO_4_ nanoparticles with *x* = 0.20 under different sintering conditions (**a**,**b**) sintering condition of 1073 K for 180 min (**c**,**d**) sintering condition of 973 K for 180 min.

**Figure 3 nanomaterials-11-03450-f003:**
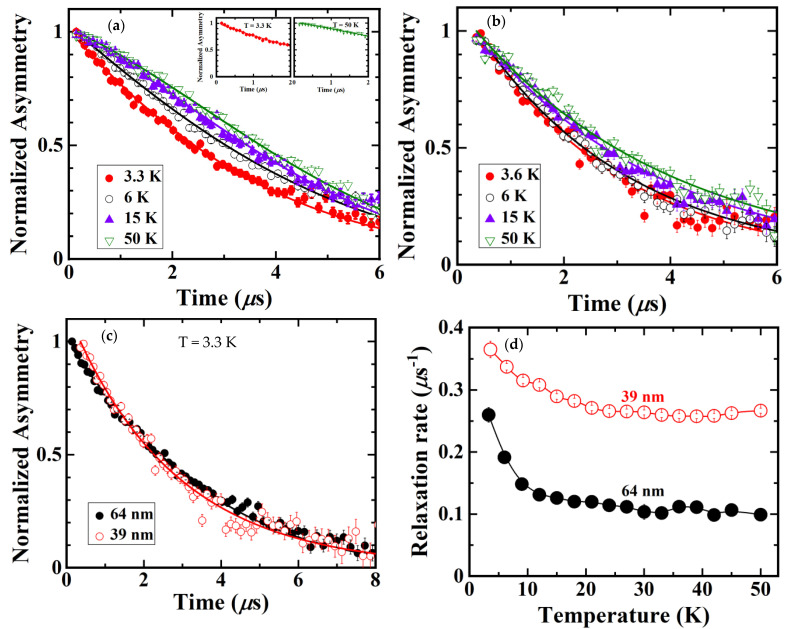
(**a**) Zero-field *μ*SR time spectra of La_2__−*x*_Sr*_x_*CuO_4_ nanoparticles with *x* = 0.20 with the particle size of 64 nm, (**b**) zero-field *μ*SR time spectra of La_2__−*x*_Sr*_x_*CuO_4_ nanoparticles with *x* = 0.20 with the particle size of 39 nm, (**c**) time dependence of normalized asymmetry for both sample at lowest measurement temperature around 3 K, (**d**) temperature dependence of muon-spin relaxation rate of La_2__−*x*_Sr*_x_*CuO_4_ nanoparticles with *x* = 0.20.

**Table 1 nanomaterials-11-03450-t001:** Rietveld refinement structural parameters for La_2__−*x*_Sr*_x_*CuO_4_ nanoparticles with *x* = 0.20.

Parameter	Sintering Condition	
	1073 K for 180 min	973 K for 180 min
Crystal structure	Tetragonal	Tetragonal
Space Group	*I4*/*mmm*	*I4*/*mmm*
Lattice constant *a* = *b* (Å)	3.7722	3.7819
Lattice constant *c* (Å)	13.2295	13.2086
Volume of unit cell (Å^3^)	188.249	188.923
Average crystallite size (nm)	64	39

**Table 2 nanomaterials-11-03450-t002:** Atomic coordinate of La_2__−*x*_Sr*_x_*CuO_4_ nanoparticles with *x* = 0.20.

Atom	Wyckoff Position	Site Symmetry	*x*	*y*	*z*	Site Occupation Factor
La1	4e	4 mm (z)	0	0	0.3609 (4)	0.9
Sr1	4e	4 mm (z)	0	0	0.3609 (4)	0.1
Cu1	2a	4/mmm (z)	0	0	0	1.0
O1	4c	mmm	0	1/2	0	1.0
O2	4e	4 mm (z)	0	0	0.184 (4)	1.0

**Table 3 nanomaterials-11-03450-t003:** Selected bond distances for La_2__−*x*_Sr*_x_*CuO_4_ nanoparticles with *x* = 0.20.

Name of Sample	Bond Distances (Å)		
	La1|Sr1–O2	Cu1–O1	Cu1–O2
Sample Reference [32]	2.3351	1.8896	2.4288
64 nm	2.3402	1.8861	2.4342
39 nm	2.3366	1.8910	2.4304

## Data Availability

Not applicable.

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
