# Peer review of "Observation of Cu Spin Fluctuations in High-Tc Cuprate Superconductor Nanoparticles Investigated by Muon Spin Relaxation"

_nanomaterials, 2021, doi:10.3390/nano11123450_

Round 1
Reviewer 1 Report
The authors present an interesting investigation of the Cu spin fluctuations in NP’s of high-Tc cuprate superconductor La2-xSrxCuO4 with x = 0.20 by means of Muon Spin Relaxation. Zero field muon spin relaxation time spectra for two sets of NP’s (64 and 39 nm) as a function of T has allowed to study the appearance of Cu spin fluctuations. At low T the observation of time decay from Gaussian to exponential is indicating a slowing down of Cu spin fluctuations. This slowing down is revealing the presence of magnetic correlations, which are not present in bulk state. No spin precession is observed, indicating the lack of long-range order. Magnetic correlation is found to be stronger for the smaller NP’s
Previous studies in the literature have demonstrated that suppression of superconductivity and enhancement of ferromagnetism is observed simultaneously with decreasing particle size in High T superconductors (HTSC). In this sense, the present study is contributing to the study of nano-size effects in HTSC. However, the study is not systematic enough to provide sound conclusions, showing results for only two sets of NP’s, which is insufficient. Furthermore, the presence of magnetic correlations must be supported and endorsed by magnetic susceptibility measurements as a function of temperature where the suppression of the Tc and the appearance of an up- turn in magnetic susceptibility have to correlate with obtained strength in magnetic correlation.
For this main reason I consider that the results, although interesting, need to be complemented with more experiments with additional NP’s sizes to show a systematic trend, and supported by magnetization studies where the effect on the suppression of the superconductivity and the appearance of magnetic correlations are shown. My recommendation is therefore ‘reconsider after major revision’.
Additionally, the authors are extrapolating the results to some conclusions which are not fully inferred from the present study. In particular the following paragraph:
‘We proposed some possible reasons to explain the appearance of magnetism in La2-xSrxCuO4 nanoparticles with x = 0.20. The crystal disruption inferred from the change in the bond distance of Cu and O might cause the charge redistribution. The charge redistribution might affect the local exchange interaction between Cu and O, leading to the appearance of magnetism. Nanoscale pinning sites is also needed to be considered as the origin of magnetism. These results have concluded that the nano-size effects might induce Cu spin fluctuations, charge redistribution, and pinning sites providing the appearance of magnetism. ‘
This part presents some speculations and could be part of the discussion, but it is not a main conclusion of the presented results.
Other minor shortcomings are:
- In the introduction the authors state that in reference 16 the studies of nano-size effects in HTSC used samples with particle sizes larger than 100 nm. In ref. 16 they use sample sizes down to 66 nm, and they observed size effects.
- The authors refer to their own previous works, ref 31 and 32, where the NP’s are analysed in detail and typical size distribution obtained from the SEM images is obtained. This information, the size distribution for the nanoparticles, must be provided as it is an important parameter.
Reviewer 2 Report
The paper reports about synthesis of La2-xSrxCuO4 nanoparticles and their investigations by XRD and mSR. The lattice parameters are in agreement with the data reported elsewhere for single crystals. The relaxation of muon-spin rapidly increases below about 10 K, indicating magnetic correlation. It is an interesting result, differing nanosize systems from crystalline ones and the paper may be recommended for publication.
However this main result was explained by the change in the bond distance of Cu and O, what looks quite cursory.
“The charge redistribution might affect the local exchange interaction between Cu and O, 262
leading to the appearance of magnetism”. – It is not the local, but two-center exchange. Also , it is more appropriate to speak about exchange part of Coulomb interaction, which may cause ferromagnetic or antiferromagnetic orientations of spins. Such speculations about the origin of the observed effect are not necessary for experimental work.
Also in some cases grammar is not clear.
- was reported to observe in gold -grammar is not clear, perhaps to be obseved
- this study remains a question perhaps the question remains in this study
- 3. it behaves superconducting below Tc. It is a trivial assertion. Tc should be indicated
- was considered to explain the origin of the 1/8 anomaly in the bulk HTSC what is the 1/8 anomaly?
- because Citric acid - why capital letter ?
Round 2
Reviewer 1 Report
I am satisfied with the response of the authors and the changes made to the manuscript. All concerns and comments are satisfactorily addressed and the work is suitable for publication in its current form.